# Perfectionistic Children and Their Parents: Is There Room for an Intergenerational Transmission? A Study of a Clinical Sample of Italian Children and Their Parents

**DOI:** 10.3390/children10030460

**Published:** 2023-02-26

**Authors:** Melania Martucci, Maria Castellano Visaggi, Giorgia Di Iorio, Mariacarolina Vacca, Valerio Zaccaria, Ignazio Ardizzone, Caterina Lombardo

**Affiliations:** 1Child Neuropsychiatry Unit, Department of Human Neuroscience, Sapienza University of Rome, 00185 Roma, Italy; 2Department of Psychology, Sapienza University of Rome, 00185 Roma, Italy

**Keywords:** perfectionism, children, parents

## Abstract

Introduction: The relationship between maladaptive perfectionism and Internalizing (ID) and Externalizing Disorders (ED) in children needs to be better understood, along with the intergenerational transmission of these traits from parents to children. The present work aimed to share light on both these issues. Method: 39 children with ID, 19 with ED, and their parents were recruited*. The Multidimensional Perfectionism Scale, the Big Five Inventory, the Child and Adolescent Perfectionism Scale, and the Hierarchical Personality Inventory for Children were used. The association between parent personality and perfectionism traits and children’s perfectionism dimensions was evaluated through hierarchical regression analysis. Results: ID and ED groups did not differ significantly concerning perfectionism. Fathers presented higher scores than mothers in Self-Oriented Perfectionism, Other-Oriented Perfectionism, and Socially-Prescribed Perfectionism. Parents of children with ID report higher levels of Obedience in their children, while parents of children with ED reported higher levels of Creativity and Perseverance. Significant associations were found between perfectionism in parents and their children, as well as between perfectionism and the personality of children. Conclusion: Results suggest a transdiagnostic nature of Perfectionism and support the transgenerational transmission of the personality traits investigated.

## 1. Introduction

Perfectionism has been traditionally described as a striving for flawlessness, involving the pursuit of self-improvement and self-demand, with a desire for order and organization despite adverse consequences, often combined with excessive critical self-evaluations [1]. One of the most influential multidimensional conceptualizations of perfectionism is the Hewitt and Flett’s model [2], which includes three different dimensions: Self-Oriented Perfectionism (SOP), which involves striving for excellence and a high standard of performance; Other-Oriented Perfectionism (OOP), referring to the tendency to set excessive standards for significant others; Socially Prescribed Perfectionism (SPP), which refers to the perception of social pressures for being perfect and the need to satisfy the high perceived expectations of others [2]. A positive association between adaptive perfectionism (e.g., SOP) and satisfaction with life, psychological well-being, and self-esteem is reported, while maladaptive perfectionism (e.g., SPP) has been associated with hopelessness and low levels of self-esteem [3,4,5,6]. Perfectionism is considered a “transdiagnostic” risk factor for developing psychopathology. Previous findings support the existence of a relationship between maladaptive perfectionism and many internalizing disorders (ID), such as depression [4,7,8,9], suicidal behavior [10], eating disorders [11,12,13], obsessive–compulsive disorder [14,15], and anxiety disorders [13,16,17]. However, it is not clear whether a similar transdiagnostic role may be found for externalizing disorders (ED), although recent findings have outlined the co-occurrence of perfectionism and impulsivity, interpreted as the reason that accounts for comorbidity across internalizing and externalizing disorders [18].

The development of perfectionistic tendencies seems to be due to both genetic and environmental influences. The genetic contribution to the intergenerational transmission of perfectionism has been reported by some authors. A Spanish study [19] showed that SPP presents a heritability of 39% in boys and 42% in girls, respectively, whereas SOP presents a heritability of 23% in boys and 30% in girls, respectively. From another study [20], it emerged that some aspects of perfectionism, such as personal standards, seem to be more heavily affected by genetics, whereas others, such as doubts about actions, might be more dependent on the environment. In general, maladaptive perfectionists are highly dependent on the responses they generate in others and worry excessively about their own failures [10,21]. Different empirical findings suggest that a child’s perfectionism develops from the desire for approval and affection from potentially demanding, controlling, and critical parents [22]. This type of dysfunction could be related to the exaggerated expectations of parents regarding the performance of their children [10]. 

Previous findings support this association providing evidence that the parents’ SOP and SPP explained a significant portion of the variance in the children’s SOP and SPP, respectively. In contrast, small positive relationships were found between the parents’ OOP and children’s OOP. According to the primary caregiver hypothesis, however, it is possible to hypothesize that the parent who happens to be mainly responsible for the care should also be mainly responsible for the development of their children’s perfectionism [23]. Since in Italy the primary caregiver is the mother, due to the longer period she usually spends with her offspring, her influence may be hypothesized as significant. 

Authors speculated that the tendency to set high standards for others might be transmitted from parent to child through other developmental intervening factors [24]. A limited number of studies have investigated the relationship between other parental factors and children’s perfectionistic traits. In the scientific literature, such factors mainly include learning models of transmission (e.g., parent modeling), parent psychopathology (e.g., parental anxiety), and parenting style (e.g., criticism/lack of warmth) [25], whereas the role of parental personality is still not widely understood. Therefore, examining the contribution of parental perfectionism and personality traits to children’s perfectionistic facets is essential, as well as considering the child’s personality characteristics that may interact with the parent’s behavior models. To this issue, previous studies indicate that certain traits, such as neuroticism [26], low extraversion [27], low agreeableness [27], and low conscientiousness [28], are associated with maladaptive perfectionism in adults. Meta-analytic evidence indicated that all the personality tendencies mentioned above are distinctive characteristics of adults with high SPP, whereas high conscientiousness is typical of individuals with high SOP [29]. 

Finally, since the development of perfectionism may be related to the personality characteristics of the children, as reported by several findings, it is important to also take into account those characteristics to shed light on the intrapersonal and interpersonal factors that promote perfectionism. For instance, Vicent and colleagues [30] oriented their focus on childhood, corroborating, for example, the hypothesis that maladaptive perfectionism could be predicted in children by their maladaptive personality traits (e.g., neuroticism and negative affect) or by the lack of adaptive personality traits (e.g., agreeableness and openness to experience) while Oros and colleagues [31] evidenced that perfectionism could be predicted in children by both their personality traits and the excessive parental demands. 

The present work aims to examine perfectionism and personality in a sample of children with ID and ED and their parents, also evaluating the intergenerational transmission of perfectionism. The specific aims of this investigation are to: -Examine the hypothesis of a transdiagnostic role of perfectionism assessing the existence of differences in perfectionistic traits between the ID and the ED groups of patients; -Explore the associations between perfectionistic traits in parents (mothers and fathers independently) and perfectionistic traits in their children.

As regards the first aim, we hypothesized that since the relationship between ID and maladaptive perfectionism in the existent literature is stronger, maladaptive perfectionism in ID children would be greater than in ED children.

As regards the second aim, consistent with the scientific literature previously summarized, we tested the following hypotheses:

Maladaptive personality traits of the children are significantly associated with their maladaptive perfectionism;

Maladaptive personality traits and maladaptive perfectionistic traits of the mothers are significantly associated with maladaptive perfectionistic traits of their children. No hypotheses were advanced for the fathers that may or may not be associated with the children’s perfectionism;

A combination of maladaptive personality and perfectionistic traits of the mother significantly predicts the maladaptive perfectionism of the children. 

## 2. Materials and Methods

### 2.1. Participants

Participants were 58 children with ID (n = 39; 53.84% males; M_age_ = 11 ± 1.57) and with ED (n = 19; 73.68% males; M_age_ = 10.84 ± 1.86) and their parents (N = 103; F = 54; M_age_ = 46.36, SD = 4.93). Children with ID presented depression (n = 15), anxiety (n = 11), comorbidity between anxiety and depression (n = 12), and obsessive-compulsive disorder (n = 1). 

Children with ED presented disruptive mood dysregulation disorder (n = 8), conduct disorder (n = 6), attention deficit hyperactivity disorder (n = 3), and oppositional defiant disorder (n = 2). All participants received their diagnoses in 2021.

### 2.2. Procedure

The drafting of this cohort study was realized according to the guidelines for reporting observational studies: Strengthening the Reporting of Observational Studies in Epidemiology (STROBE) Statement.

The study design was cross-sectional. Children were recruited at the Department of Human Neurosciences—Child Neuropsychiatry Unit, University of Rome “Sapienza” from January 2021 to December 2021. The study was conducted in accordance with the Declaration of Helsinki and approved by the Ethical Committee of the Department of Psychology (Prot. n. 0000858)”.

Study participants were enrolled based on the following inclusion criteria: 

Children aged between 7 and 14 years and their parents;

Diagnosis of internalized or externalized disorder, established according to the DSM-5 criteria [32] and backed up using the semi-structured interview Schedule for Affective Disorders and Schizophrenia for School-Age Children-Present and Lifetime Version (K-SADS- PL) [33], a useful tool for the assessment of psychopathology in children, which was administered to parents by an experienced child psychiatrist;

Parental written informed permission to participate.

Exclusion criteria were:

A diagnosis of intellectual disability in the child;

Any neurological or neurosensorial diseases in the child or their parents.

At the first assessment visit, parents and children were informed about the study and invited to participate. All those who accepted were tested with the instruments described below.

### 2.3. Instruments

Two groups of questionnaires were used, one assessing the characteristics of the child and one assessing the characteristics of their parents. Questionnaires were filled in either by the child or by the parents, as specified below.

The Child assessment consists of the following:

Child and Adolescents Perfectionism Scale (CAPS) [34] filled in by the child: it is an adaptation of the Hewitt and Flett Multidimensional Perfectionism Scale (MPS), used with adult populations. In its original form [34], the CAPS consisted of 22 items intended to measure SOP and SPP. We used the Italian version [35] of the abbreviated 14-item version [36]. This scale measures SPP (7 items, e.g., “There are people in my life who expect me to be perfect”) and two aspects of SOP, as previous factor analytic studies [36] with samples of children and adolescents have indicated that SOP split into two facets. These aspects have been named as SOP-Critical (4 items, e.g., “I get mad at myself when I make a mistake”) and SOP-Striving (3 items, e.g., “I try to be perfect in everything I do”) and respectively assess the propensity to engage in excessive self-criticism and to set high standards of performance [36]. Participants rated each statement on a 5-point Likert scale (from 1 = not at all true of me to 5 = very true of me). The original validated version of the CAPS showed good psychometric properties (Cronbach’s alpha ranging from 0.65 to 0.81) [35]. Internal reliability coefficients computed on the present sample resulted equal to 0.79, 0.67, and 0.71, respectively, for SPP, SOP-Critical, and SOP-Striving; 

The Hierarchical Personality Inventory for Children [37] filled by parents: HiPIC is organized hierarchically, allowing to define a personality profile with different levels of generality through only 108 items, providing a detailed description of the child’s personality profile regarding the five general domains (i.e., Extraversion, Benevolence, Conscientiousness, Emotional Stability, and Imagination) and the 18 individual facets (Extraversion: Energy, Expressiveness, Optimism, Shyness; Benevolence: Altruism, Dominance, Egocentrism, Compliance, Irritability; Conscientiousness: Concentration, Perseverance, Order, Achievement striving; Emotional Stability: Anxiety, Self-confidence; Imagination: Creativity, Intellect, Curiosity). Cronbach’s alphas for the HiPIC dimensions, as indicated in the Italian-validated version of the scale, ranged from 0.65 to 0.93 [Eisinga 2013]. In the present sample, we found alpha coefficients of the five domains ranging from 0.60 to 0.911. High scores indicated high levels of the traits.

The parent’s assessment includes the following:

Multidimensional Perfectionism Scale—Short version (MPS-S) [38] consists of a total of 15 questions, with three subscales—each comprising five items, namely SOP (e.g., “One of my goals is to be perfect in everything I do”), SPP (e.g., “The better I do, the better I am expected to do”) and OOP (e.g., “I have high expectations for the people who are important to me”). The items are scored on a Likert scale from 1 (strongly disagree) to 7 (strongly agree), with higher scores indicating greater perfectionistic tendencies. Cronbach’s alphas for the Italian MPS-B ranged from 0.70 to 0.86. The reliability coefficients for the present sample indicated similar values suggesting good internal consistency for each of the three subscales (i.e., SOP: α = 0.88; SPP: α = 0.83; OOP: α = 0.76);

Big Five Inventory (BFQ) [39] is a ten-item scale designed to assess the Big Five dimensions in a very short amount of time. The Italian version showed good psychometric properties (Guido et al., 2015). The Spearman–Brown formula for the reliability estimate of the five scales was within the acceptable range (0.2–0.4) as suggested by Briggs and Cheek (1986) [40]. High scores indicated greater levels of the traits.

### 2.4. Statistical Analysis

The normal distribution of the data was assessed through the Shapiro-Wilk test. To minimize cumulative type-1 error, the significance level was set at *p* < 0.01. Cronbach’s alphas were calculated for each questionnaire as a measure of reliability, with α of 0.6—0.7 indicating an acceptable level of internal consistency [41]. One exception was the BFI, which contains 10 latent constructs measured by only two items. For this scale, we used the Spearman–Brown formula, which is more appropriate for the reliability estimate of scales with less than three indicators [38]. The acceptable range (0.2–0.4) suggested by Briggs and Cheek [42] was used to estimate the reliability of the BFI. 

Multivariate analysis of variance (MANOVA) was used for comparing children with ED and children with ED as well as the respective groups of parents on personality dimensions and perfectionism aspects. MANOVA conducted on parents was performed controlling for their gender (mothers vs. fathers). Correlation analyses were conducted to estimate intercorrelations, the associations between parent perfectionism (i.e., MPS-B) and children perfectionism (i.e., CAPS-14), as well as the associations between perfectionistic aspects of children (i.e., CAPS-14), and their personality reported by parents (i.e., HiPIC). 

Three separate hierarchical regression analyses were performed to identify whether the mother’s and father’s perfectionism (Step 1) and Big Five dimensions (Step 2) predicted each of the three perfectionistic facets (SPP, SOP-Critical, SOP-striving) of their children. Analyses were conducted using IBM SPSS Statistics (Version 27).

## 3. Results

### 3.1. Descriptive Statistics

Mean scores and standard deviations for each aspect explored in children and their parents are displayed in Table 1 and Table 2. Appendix A.

### 3.2. Perfectionism Self-Reported by ID and ED Children

A MANOVA comparing CAPS scales of children with ID and ED was performed. Results evidenced that both the main effect and the univariate effects were non-significant differences in perfectionistic aspects between the two groups (*p* > 0.05; partial η^2^ = 0.03).

Differences between ED and ID personalities of children referred by their parents:

To examine differences between ED and ID children’s personalities referred by their parents, a MANOVA was computed considering parent groups (ED vs. ID) as independent factors and controlling for gender (mothers vs. fathers). Results evidenced that parents of children with ED reported higher levels of Creativity (M = 4.057; SD = 2.209; F(_1, 100)_ = 4.184, *p* < 0.05; partial η^2^ = 0.04) and Perseverance (M = 4.80; SD = 2.083; F_(1, 100)_ = 6.116, *p* < 0.05; partial η^2^ = 0.06) for their children, compared to parents of children with ID (respectively, M = 3.117; SD = 2.175 and M = 3.794; DS = 1.897). Moreover, parents of children with ID reported higher levels of Obedience (M = 4.853; SD = 1.668; F_(1, 100)_ = 5.422, *p* < 0.05; partial η^2^ = 0.05) in their children, compared with parents of children with ED (M = 3.137; SD = 1.141). No effects were observed for gender. 

### 3.3. Personality Characteristics and Perfectionism of the Parents

To examine perfectionism and personality in ED and ID parents, a MANOVA was used to compare MPS-S and BFI scores reported by these two groups, controlling for gender (mothers vs. fathers). Results indicated no significant effect for the group of diagnosis. However, a significant effect of gender was observed, with fathers presenting higher scores in MPS-SOP (M = 21.816; SD = 7.661; F_(1, 100)_ = 5.207, *p* < 0.05; partial η^2^ = 0.05), higher MPS-OOP (M = 19.897, SD = 5.986; F(_1, 100)_ = 7.561, *p* < 0.01; partial η^2^ = 0.07), and higher MPS-SPP (M = 17.795, SD = 7.424; F_(1, 100)_ = 4.712, *p* < 0.05; partial η^2^ = 0.05) than mothers (SOP: M = 18.333, SD = 7.541; OOP: M = 16,333, SD = 6.796; SPP: M = 14.759, SD = 6.564).

Correlation analyses. The bivariate associations between perfectionistic aspects of parents and children:

Intercorrelations between CAPS-14 subscales were all statistically significant, ranging from r = 0.37 (*p* < 0.01) to r = 60 (*p* < 0.01). Similar results were found for the MPS subscales assessing perfectionism in fathers and the MPS subscales assessing perfectionism in mothers, with correlation coefficients ranging from r = 0.55 (*p* < 0.01) to r = 69 (*p* < 0.01). 

Evaluating the relationships between parent and children characteristics, significant correlations were observed between SOP-Critical (CAPS) and MPS-OOP (r = 0.30, *p* < 0.05) as well as MPS-SPP (r = 0.30, *p* < 0.05), between SOP-Striving (CAPS) and MPS-SOP (r = 0.29, *p* < 0.05) and SPP (r = 0.33, *p* < 0.05). Considering the perfectionistic dimensions in mothers, MPS-SOP (r = 0.28, *p* < 0.05) and MPS-SPP (r = 0.33, *p* < 0.05) were significantly related to SOP-Striving (CAPS). Moreover, MPS-OOP (r = 0.30, *p* < 0.05) and SPP (r = 0.40, *p* < 0.01). In fathers, MPS-SPP was significantly related to SOP-Striving (CAPS) (r = 0.29, *p* < 0.05), SOP-Critical (CAPS) (r = 0.43, *p* < 0.01), and SPP-CAPS (r = 0.38, *p* < 0.01). Finally, MPS-SOP and MPS-OOP (rs= 0.34, *p* < 0.01) were significantly related to SOP-Critical (CAPS).

Correlation analyses. The bivariate associations between perfectionistic aspects of children and their personality:

Positive and significant associations were found between SOP-Critical (CAPS) and conscientiousness (r = 0.26, *p* < 0.01), Anxiety (r = 0.22, *p* < 0.05), Creativity (r = 0.21, *p* < 0.05), Dominance (r = 0.265, *p* < 0.01), Egocentrism (r = 0.26, *p* < 0.01), and Achievement striving (r = 0.31, *p* < 0.01). SOP-Striving (CAPS) was significantly related to higher Imagination (r = 0.29, *p* < 0.01), Benevolence (r = 0.24, *p* < 0.05), Anxiety (r = 0.26, *p* < 0.01), Shyness (r = 0.23, *p* < 0.05), Intellect (r = 0.32, *p* < 0.01), Altruism (r = 0.22, *p* < 0.05), and Achievement striving (r = 0.20, *p* < 0.05). Finally, one significant association was observed between SPP (CAPS) and Egocentrism (r = 0.25, *p* < 0.05).

Hierarchical Regression. Do Perfectionism and the Personality of the parents predict children’s perfectionistic facets?

Regression analyses performed on the mother’s reports showed that variables entered in Step 1 explained the 12% of the variance of SOP-Critical (R = 0.41, F_(3, 53)_ = 3.36, *p* < 0.05), with the mother’s MPS-SPP (ß = 0.388, *p* < 0.05) positively predicting children’s SOP-Critical.

Variables entered in Step 2 accounted for 37% of the variance of SOP-Critical (R = 0.68, F_(3, 53)_ = 4.86, *p* < 0.001). More specifically, the mother’s MPS-SPP (ß = 0.456, *p* < 0.01), Agreeableness (ß = 0.475, *p* < 0.001), Conscientiousness (ß = −0.310, *p* < 0.05), and Openness (ß = 0.272, *p* < 0.05) uniquely predicted children’s SOP-Critical. Lastly, regression analyses performed on fathers indicated that variables entered in Step 1 explained 10% of the variance of CAPS-SPP (R = 0.39, F_(3, 48)_ = 2.82, *p* < 0.05), with the father’s MPS-SPP (ß = 0.434, *p* < 0.05) positively predicting children’s CAPS-SPP. Lastly, variables entered in Step 2 explained the 19% of the variance of CAPS-SPP (R = 0.57, F_(3, 48)_ = 2.44, *p* < 0.05) and showed that Agreeableness (ß = −0.355, *p* < 0.05), Conscientiousness (ß = −0.341, *p* < 0.05) and MPS-SPP (ß = 0.402, *p* = 0.05) uniquely contributed to children’s CAPS-SPP. No other regression evidenced statistically significant effects. 

## 4. Discussion and Conclusions

The present study aimed to shed light on which variables explain the presence and intensity of perfectionism in a clinical sample of children with ID or ED and their parents. The comparison of perfectionism between ID and ED children evidenced no statistically significant differences. This finding is consistent with previous scientific literature suggesting that perfectionism is related to a great number of psychopathological syndromes such as alcoholism, anorexia, depression, and personality disorders but also conduct problems, disruptive behavior (difficulties), and hyperactivity [2], thus suggesting a potential role of perfectionism as a transdiagnostic risk factor also for ED. 

Regarding the association of personality dimensions and perfectionism of the children, our results confirmed that conscientiousness was significantly related to SOP-Critical (adaptive). Moreover, a positive association between Anxiety (Emotional Stability), Creativity (Imagination), Dominance (Benevolence), Egocentrism (Benevolence), Achievement striving (Conscientiousness), and SOP Critical (adaptive) was found. SPP (maladaptive) had a significant positive association with Egocentrism. Hence, Egocentrism, which is a facet of the general domain of Benevolence, could have a double meaning in the developmental trajectory of the children’s perfectionism. Dunkley et al. 2006b [27] found a positive relation between self-critical perfectionism (maladaptive), neuroticism, and depression and a negative relation between extraversion and agreeableness. Furthermore, there is a consistent association between socially prescribed perfectionism (maladaptive), higher neuroticism [43], and depression [28], while self-oriented perfectionism (adaptive) is related to conscientiousness [28]. Lahey and Waldman suggested that anxiety symptoms would negatively correlate with novelty seeking because novelty seeking is inversely related to shyness or behavioral inhibition [44]. Our study suggests that ED children present higher levels of creativity which is part of the general domain of Imagination and irritability, part of Benevolence. Creativity, in particular, stimulates the research of new brilliant ideas correlated with novelty seeking. Meanwhile, irritability is a typical characteristic of externalizing disorders, considering that [45] the core psychological feature of diverse common forms of psychopathology is distress or negative emotionality, and the core feature of externalizing forms of psychopathology is disinhibited distress. The use of the term ‘externalizing’ also explicitly links the child and adult literature, which both reveal the existence of a coherent group of disinhibitory disorders that lead to externalized forms of distress [46]. ID children are described as Obedient. Obedience (a facet of Benevolence) could be related to shyness and inhibition, typical of anxiety and depression; therefore, ID children are at risk of being undiagnosed. 

The last aim of the study was the assessment of the eventual transgenerational transmission of perfectionism. Before discussing the relationships between parents and children characteristics, some comments could be proposed regarding the gender differences found in the mother’s and father’s perfectionism: fathers reported higher scores in MPS-SOP, MPS-OOP, and MPS-SPP than mothers, consistent with previous findings [47]. 

Although research on the intergenerational transmission of perfectionism is still emerging, some findings have led to considerable insights. For example, Cook and Kearney [48] observed that mothers who perceived others as having unreasonably high expectations (i.e., SPP) were more prone to have children who pursued self-oriented perfectionistic standards of performance (i.e., SOP) and who perceived significant others as excessively demanding (i.e., SPP). These results suggest that mothers with high SPP may contribute to their children’s SOP and SPP by attaching irrational importance to pleasing and impressing others and being susceptible to harsh self-criticism. Moreover, the authors found that the mother’s OOP positively predicted their child’s SPP and SOP, whereas no significant associations were detected between the father’s and children’s perfectionism. This last finding aligns with a previous study [49] evidencing the mother’s but not the father’s general perfectionism as a significant predictor of daughters’ perfectionism. Smith and colleagues [50] conducted a multi-source investigation into the father’s OOP and daughter’s perfectionism, evaluating the self-critical and the personal standards components of the trait. The authors reported that the father’s OOP uniquely predicted self-critical and personal standards aspects of perfectionism in daughters, suggesting that daughters with fathers who demand perfection could develop both adaptive and maladaptive perfectionistic sides. More recently, a meta-analysis [24] addressed the extent to which parental expectations and criticism contribute to offsprings’ SOP, OOP, and SPP.

In our study, the correlation analyses showed a statistically significant relationship between parent’s and children’s perfectionism. However, if perfectionism is to be fully examined, it is necessary to take a contextual approach, where parents, personality, and culture all have a part to play in its development. Actually, the mother’s socially prescribed perfectionism, together with their agreeableness, conscientiousness, and openness, predict high personal standards (adaptive perfectionism) in their children. On the contrary, the father’s Agreeableness, Conscientiousness, and MPS-SPP explained the children’s critical perfectionism (CAPS-SPP).

These findings suggest not only that the perfectionism of the parents may promote the development of perfectionistic tendencies in children but also that the influence may be different for mothers and fathers. Mothers, the principal childcare providers in Italy, seem to influence the perfectionism of children, increasing positive striving, while the father’s socially prescribed perfectionism seems to promote children’s dysfunctional perfectionistic tendencies. This is relevant data if we consider that fathers showed higher scores than mothers in all the dimensions of perfectionism. However, differential influences of mothers and fathers may be hypothesized on female and male children, consistent with the Social Expectations Model [1], which posits that children tend to imitate perfectionism exhibited by their same-gender parents. The small sample size prevented us from testing this differential influence. 

Hence, the personality dimensions described in the literature associated with individual perfectionism (Parental Conscientiousness and Agreeableness associated with child SOP-critical) are associated with the same parental personality dimensions. If confirmed by future studies, the present findings confirm the transgenerational transmission of perfectionism from parents to children with a different role for mothers and fathers. 

## 5. Limitations and Future Perspective

Before concluding, some limitations should also be acknowledged. 

First, the sample size is small; thus, replication studies are necessary for generalizing our findings. Moreover, we examined a clinical sample, and thus, results should not be extended to the general population. Future perspectives could include more extensive clinical samples to determine the role of perfectionism in the rise of psychopathology. In this regard, recent evidence is trying to find a p-factor, considered an individual’s overall predisposition toward psychopathology. Referring to internalizing disorders, deficits in the updating, in the working memory, and in the shifting mental set seem predisposed to depression and anxiety. Instead, shifting mental sets and working memory deficits are involved in developing externalizing disorders [51,52,53]. 

In the future, it could be interesting to analyze these factors in association with perfectionism in our clinical sample.

Another limitation is using parental reports of children’s personalities that may be biased (e.g., some parents may have under-reported their children’s problematic personality attitudes), although they provide one external source of information that could be integrated with multimethod assessments in future studies.

## Figures and Tables

**Table 1 children-10-00460-t001:** Children’s perfectionism scores.

CAPS-14	Total Sample (M, SD)	ID Children (M, SD)	ED Children (M, SD)
SOP-Critical	2.27 (0.98) Z = −3.43	2.26 (1.01) Z = 1.78	2.29 (0.94) Z = 1.39
SOP-Striving	3.08 (1.14) Z = −1.24	3.08 (1.20) Z = −1.02	3.10 (1.04) Z = −0.63
SPP	2.63 (0.90) Z = −4.39	2.66 (0.95) Z = −3.36	2.59 (0.79) Z = −2.74

Note. Z: Zeta scores; CAPS-14: Child and Adolescents Perfectionism Scale (self-reported); SOP: Self-Oriented Perfectionism; SPP: Socially Prescribed Perfectionism; ID: internalizing disorders; ED: externalizing disorders.

**Table 2 children-10-00460-t002:** Descriptive of parent’s self-reported perfectionism and personality (MPS-S, BFI-10) and parent’s reported personality of the children (HiPIC—five dimensions) ^1^.

Variable	ID Mothers (N = 37) M (SD)	ED Mothers (N = 17) M (SD)	ID Fathers (N = 31) M (SD)	ED Fathers (N = 18) M (SD)
MPS-SOP	18.67 (6.88) Z = 1.99	17.58 (8.99) Z = 0.70	20.77 (7.81) Z = 3.50	23.61 (7.25) Z = 4.40
MPS-OOP	16.22 (6.38) Z = 1.62	16.58 (7.83) Z = 1.39	18.93 (6.03) Z = 4.44	21.55 (5.67) Z = 5.57
MPS-SPP	14.78 (6.21) Z = 0.35	14.70 (7.44) Z = 0.18	17.38 (6.83) Z = 3.05	18.50 (8.50) Z = 3.22
BFI-Agreeableness	3.42 (1.02) Z = −4.59	3.67 (0.61) Z = −2.29	3.34 (.83) Z = −4.57	3.25 (1.04) Z = −3.79
BFI-Conscientiousness	4.21 (0.71) Z = −0.21	4.29 (0.68) Z = 0.08	4.08 (0.86) Z = −0.70	4.16 (0.72) Z = −0.29
BFI-Extroversion	3.54 (0.87) Z = −4.35	3.38 (0.96) Z = −3.44	3.27 (0.91) Z = −5.11	3.0 (0.82) Z = −4.75
BFI-Openness	3.24 (1.01) Z = −5.82	3.11 (1.22) Z = −4.33	2.98 (0.81) Z = −6.36	3.47 (1.03) Z = −3.35
BFI-Emotional Stability	3.27 (1.05) Z = −3.12	3.23 (1.01) Z = −2.23	3.35 (0.95) Z = −2.55	3.69 (0.71) Z = −0.96
HiPIC-Emotional Stability	3.19 (1.37)	3.35 (1.80)	3.36 (1.56)	2.61 (1.50)
HiPIC-Extraversion	3.24 (1.51)	3.18 (1.30)	3.35 (1.70)	3.44 (1.94)
HiPIC-Imagination	3.14 (1.94)	3.41 (1.87)	3.61 (1.99)	3.83 (1.69)
HiPIC-Benevolence	4.05 (2.08)	3.35 (1.49)	3.23 (1.84)	3.17 (2.12)
HiPIC-Conscientiousness	3.59 (0.92)	3.53 (0.71)	3.68 (1.35)	3.22 (0.87)

Note. Z: Zeta scores: MPS: Multidimensional Perfectionism Scale; SOP: Self-Oriented Perfectionism; SPP: Socially Prescribed Perfectionism; ID: internalizing disorders; ED: externalizing disorders; HiPIC: Hierarchical Personality Inventory for Children (this inventory was filled in by each parent and indicate personality characteristics of the children). ^1^ For this scale, mean Sten scores were indicated.

## Data Availability

Data is contained within the article or Appendix A.

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
