# Peer review of "Perfectionistic Children and Their Parents: Is There Room for an Intergenerational Transmission? A Study of a Clinical Sample of Italian Children and Their Parents"

_children, 2023, doi:10.3390/children10030460_

Round 1

Reviewer 1 Report

Perfectionism, Personality and Children Psychopathology: An 2 intergenerational transmission? A Study of a clinical sample of 3 Italian children and their parents.

Journal: Children

Dare: 30/01/23

Overall this study has promise but the manuscript needs substantial work. Below I detail what is required.

Abstract:

The abstract needs to be rewritten to contain the main rationale and research question/ hypotheses. And follow these with the main results and interpretations related to these. It seems disjointed and unclear.

Introduction:

The first page of the introduction needs to be clearer. It reads like a bunch of summaries of papers rather than a narrative leading to your rationale.

There is a good rationale here, but it is not being brought out by the introduction.

You need a clear research question and clear falsifiable hypotheses.

Materials and methods:

This section’s structure needs to be adjusted. It is very hard to follow.

You need to provide your ethics approval number.

Need much more demographic information on the sample including number and types of diagnosis, how long they have had the diagnosis etc etc.

Can you provide Z scores for the instruments normalised to a norm population? For instance I do not know if those scores on the SOP are high or low. 

Results:

This needs to be rewritten to neatly follow the narrative, discreetly testing each hypothesis, it is very messy in its current form and hard to follow.

You seem to have done A LOT of t tests and not done any adjustments for multiple comparisons. Why?

You need to regard a measure of effect size with each t-test.

Discussion:

This needs to be rewritten. Especially the first page. Again, it should follow the narrative from your rationale to testing your hypotheses to your interpretation of your findings with regard to the other literature. It is very hard to follow.

“In the comparison of perfectionism between ID children and ED children, no statisti-cally significant differences were found, suggesting that perfectionism may be a transdi310 agnostic risk for different psychopathologies”. Be carful saying this. It is a cross sectional sample.

“However, due to the small sample size, it is also possible that the statistical power is small and differences eventually present were not revealed by the data. It is thus possible that increasing sample size may show those differences. However, the findings is consistent with previous scientific literature suggesting that perfectionism is related to a great number of psychopathological syndromes such 315 as alcoholism, anorexia, depression, personality disorders but also conduct problems, disruptive behaviour (difficulties) and hyperactivity [1]” There are a few things wrong with this. First there are two following sentences both starting with ‘however’. Second, if you are saying it is because of your small sample size then what is the point of your study?

I would suggest a separate ‘limitations and directions for future research’ section. You need to discuss this in more detail.

“Moreover, we examined a clinical sample thus results should not be extended to the 365 general population.” Great point! ED and ID also fit neatly with a dimensional approach. I suggest reading and citing the work of Haywood et al for this sentence. Suggested papers are below:

Haywood, D., Baughman, F. D., Mullan, B. A., & Heslop, K. R. (2021). Psychopathology and Neurocognition in the Era of the p-factor: The Current Landscape and the Road Forward. Psychiatry International2(3), 233-249.

Haywood, D., Baughman, F. D., Mullan, B. A., & Heslop, K. R. (2022). What Accounts for the Factors of Psychopathology? an Investigation of the Neurocognitive Correlates of Internalising, Externalising, and the P-factor. Brain Sciences12(4), 421.

Haywood, D., Baughman, F. D., Mullan, B. A., & Heslop, K. R. (2022). Neurocognitive Artificial Neural Network Models Are Superior to Linear Models at Accounting for Dimensional Psychopathology. Brain Sciences12(8), 1060.

Author Response

Abstract:

  • R1: The abstract needs to be rewritten to contain the main rationale and research question/ hypotheses. And follow these with the main results and interpretations related to these. It seems disjointed and unclear.

A: Thank you for your comment. We rewrote the abstract to give a more clear narrative of our work. Due to word limitations (200 words allowed for the abstract) we couldn’t give more details about the hypotheses but we hope that in the present version it is more informative that in the previous version.

Introduction:

  • R1: The first page of the introduction needs to be clearer. It reads like a bunch of summaries of papers rather than a narrative leading to your rationale.There is a good rationale here, but it is not being brought out by the introduction. You need a clear research question and clear falsifiable hypotheses.

A: Thank you for your comment. We rewrote completely the introduction in order to meet the reviewer’s request. We are confident that in the present form, the introduction is more clear and the rational more evident.

Materials and methods:

  • R1: You need to provide your ethics approval number.

A: Thank you for this suggestion. The project’s ethics approval number was provided, as follows:

“The research was approved by the Ethical Committee of the Department of Psychology (Prot. n. 0000858)”

  • R1: Need much more demographic information on the sample including number and types of diagnosis, how long they have had the diagnosis etc etc.

A: We agree with this comment and have incorporated your suggestion including the following information:

“Children with ED presented depression (n = 15), anxiety (n = 11), comorbidity between anxiety and depression (n = 12), and obsessive-compulsive disorder (n = 1). Children with ID presented disruptive mood dysregulation disorder (n = 8), conduct disorder (n = 6), attention deficit hyperactivity disorder (n = 3), and oppositional defiant disorder (n = 2). All participants received their diagnoses during 2021.“

  • R1: Can you provide Z scores for the instruments normalised to a norm population? For instance I do not know if those scores on the SOP are high or low. 

A: Thank you for this point raised. We computed the Z scores for MPS-S, CAPS and BFI and indicated the relative values for each subscale. For the HiPIC’s general domains, Sten score was indicated in the original version of the manuscript, as suggested by the validation study:

DI BLAS, L., Serafino, F., & De Fruyt, F. (2005). La versione italiana del Hierarchical Personality Inventory for children (HiPIC). Eta'Evolutiva, 82, 41-53.

Results:

  • R1: You seem to have done A LOT of t tests and not done any adjustments for multiple comparisons. Why?

A: Thank you. This concern was addressed together with Reviewer 2 point 7 comment. Since, according his/her point we used a different analytical strategy, any adjustment for multiple comparisons is no more needed (see our response to reviewer’s 2 seventh point).

  • R1: You need to regard a measure of effect size with each t-test.

A: As the t-test was no longer used in the new version of the manuscript (according to reviewer’s2 comments), we provided partial η2 as a measure of effect size referred to MANOVA results. We hope that this new version of the manuscript will fulfil the reviewer’s demands.

Discussion:

R 1. : This needs to be rewritten. Especially the first page. Again, it should follow the narrative from your rationale to testing your hypotheses to your interpretation of your findings with regard to the other literature. It is very hard to follow.

“In the comparison of perfectionism between ID children and ED children, no statisti-cally significant differences were found, suggesting that perfectionism may be a transdi310 agnostic risk for different psychopathologies”. Be carful saying this. It is a cross sectional sample.

Thank you. We modified almost totally the discussion according to the reviewer’s suggestions. We hope that in the present version it meets the reviewer’s requirements.

R1: “However, due to the small sample size, it is also possible that the statistical power is small and differences eventually present were not revealed by the data. It is thus possible that increasing sample size may show those differences. However, the findings is consistent with previous scientific literature suggesting that perfectionism is related to a great number of psychopathological syndromes such 315 as alcoholism, anorexia, depression, personality disorders but also conduct problems, disruptive behaviour (difficulties) and hyperactivity [1]” There are a few things wrong with this. First there are two following sentences both starting with ‘however’. Second, if you are saying it is because of your small sample size then what is the point of your study?

I would suggest a separate ‘limitations and directions for future research’ section. You need to discuss this in more detail.

A: Thank you for the comment. In the revised version of the paper there is a section “limitations and directions for future research” where we discuss also the small sample size.  Furthermore we changed the sentence as follows:

“The comparison of perfectionism between ID and ED children evidenced no statistically significant differences.  This finding is consistent with previous scientific literature suggesting that perfectionism is related to a great number of psychopathological syndromes such as alcoholism, anorexia, depression, personality disorders but also conduct problems, disruptive behavior (difficulties) and hyperactivity [1] thus suggesting a potential role of perfectionism as a transdiagnostic risk factor also for ED.  However, due to the small sample size, it is also possible that the small statistical power prevented to reveal differences that may be small.”

R1: “Moreover, we examined a clinical sample thus results should not be extended to the 365 general population.” Great point! ED and ID also fit neatly with a dimensional approach. I suggest reading and citing the work of Haywood et al for this sentence. Suggested papers are below:

A: Thank you very much for the suggestion. We have read and cited these papers at the end of the discussion.

Haywood, D., Baughman, F. D., Mullan, B. A., & Heslop, K. R. (2021). Psychopathology and Neurocognition in the Era of the p-factor: The Current Landscape and the Road Forward. Psychiatry International2(3), 233-249.

Haywood, D., Baughman, F. D., Mullan, B. A., & Heslop, K. R. (2022). What Accounts for the Factors of Psychopathology? an Investigation of the Neurocognitive Correlates of Internalising, Externalising, and the P-factor. Brain Sciences12(4), 421.

Haywood, D., Baughman, F. D., Mullan, B. A., & Heslop, K. R. (2022). Neurocognitive Artificial Neural Network Models Are Superior to Linear Models at Accounting for Dimensional Psychopathology. Brain Sciences12(8), 1060.

Reviewer 2 Report

Review: Perfectionism, Personality and Children Psychopathology: An intergenerational transmission? A Study of a clinical sample of Italian children and their parents.

This paper is about a relevant research, on a not enough explored topic, and it gives a good contribution to the field. It is well structured and well written. Nevertheless, it has weaknesses that should be improved and aspects to clarify.

Title:

I think it is misleading because it indicates that intergenerational transmission of personality and of psychopathology is also addressed; The research addresses only the intergenerational transmission of perfectionism…

Abstract:

- “Moreover, ED group 24 fathers reported higher levels of self-oriented perfectionism and higher levels of other-oriented perfectionism than ED group mothers”.- I do not see data pertaining to this variable presented in the Results.

Introduction:

- P. 1, line 33: “Perfectionism is a personality trait involving the pursuit of self-improvement, self-demand and a desire for order and organization despite adverse consequences, combined with excessive critical self- evaluations[1]” - Hewitt and Flett do not present perfectionism as a trait, but as a personality style or a pattern involving individual and social features…

- When perfectionism and intergenerational transmission is addressed, genetics must be mentioned. At least a reference about it should be included, and again mentioned in the discussion.

- In the literature review of perfectionism and personality traits, studies with children are missing, and they should be included as children are the focus of this study. Namely, “Child Perfectionism and its Relationship with Personality, Excessive Parental Demands, Depressive Symptoms and Experience of Positive Emotions (Oros et al., 2017), “Clarifying the two facets of Self-Oriented Perfectionism: influences on affect and the Big Five traits of personality in children (Vicent et al., 2019).

- P. 3, line 92: These studies about the relation of perfectionism with personality dimensions are with adults, and it should me mentioned. After that, studies with children should be introduced.

Method:

- Pertaining to assessment of the children´s personality, why did not the authors choose a self-report instrument, and instead they chose assessment by the parents? This must be explained in the paper, and it should come in the limitations too (see below).

- P.4, linha 181,” In the present sample, we found alpha coefficients of the five domains ranging from .60 to .911, with high scores indicating high levels of the traits”. - This sentence seems to indicate that higher alfa values indicate higher trait levels…it should be reformulated.

- P.4, line 171, The Hierarchical Personality Inventory for Children: the authors should present to which five general domains the facets belong; in the same vein, in the Discussion, when facets are presented, they should be discussed as framed in the respective general domain.

Results:

- The authors mention tables SI-     , it is incomplete, and I do not see any of these tables. A table with all the correlations should be presented. 

- P.5, line 211, “Three separate hierarchical regression analyses were performed to identify whether  mothers and fathers’ perfectionism (Step 1) predicted…”. - All the steps should be indicated. In p.8, line 299, they indicate step 2 only…these informations should be uniform.

- P.7, line 245, “T-tests comparing perfectionism (i.e., MPS) and personality (i.e., BFI-10) of parents of ID and ED children did not show any significant difference between these two groups of parents (all ps > .05). T-test comparing each scale of the MPS and of the BFI-10 filled in by mothers to those filled in by fathers were conducted separating ID and ED children. Results indicated that ED fathers reported higher levels of SOP (t = 2.19, p < .05) and higher levels of OOP (t = 2.16, p < .05) than ED mothers. No other significant differences were observed”. - I do not understand this paragraph. In the first sentence the authors state that there are no differences…Then, in the second sentence they mention the same variables of the first sentence, but state something different, and in the last sentence they state there are differences…so what does the first sentence refer to?

- T-test is to one dependent variable; to multiple dependent variables Manova should be used. Or, in alternative, Bonferroni correction should be used in order to counteract the multiple comparisons problem.

- P.7, line 253, “Since evaluations of children’s personality by mothers and fathers did not differ (as reported above), the t-test comparing personality traits of diagnostic groups of children (ID and ED) were conducted on the mean scores of both parents evaluations”. - Some paragraphs below the authors report data of mother and father separately, it is not coherent. In the Abstract this separated data is also present.

Discussion:

In general, I think it this section should be improved, clearer (by discussing the personality facets framed in the respective general domain, as I stated above) and more linked with theoretical aspects presented in the Introduction.

- P.9, line 349, Agreeableness is repeated.

- Neuroticism did not have significant effect in any of the analyses; this must be addressed, as it is not expected by the literature.

- The theoretical models that explain transgenerational transmission of perfectionism, which are presented in Introduction, should be included in the Discussion, addressing the results. As an example, when the authors state “These findings suggest not only that perfectionism of the parents may promote the development of perfectionistic tendencies in children but also that the influence may be different for mothers and fathers.”, this should be related with theoretical aspects presented in the Introduction.

Limitations:

- The report of personality characteristics of the children by the parents should be here, as there may be a bias that affects the relation between characteristics of the parents and reported characteristics of the children.

Author Response

Title:

  • R2: I think it is misleading because it indicates that intergenerational transmission of personality and of psychopathology is also addressed; The research addresses only the intergenerational transmission of perfectionism…

A: thank you for your comment. We modified the title focusing on perfectionism.

Abstract:

  • “Moreover, ED group 24 fathers reported higher levels of self-oriented perfectionism and higher levels of other-oriented perfectionism than ED group mothers”.- I do not see data pertaining to this variable presented in the Results.

A: thank you. We modified the abstract so this data is no longer included.

Introduction:

  • R2: 1, line 33: “Perfectionism is a personality trait involving the pursuit of self-improvement, self-demand and a desire for order and organization despite adverse consequences, combined with excessive critical self- evaluations[1]” - Hewitt and Flett do not present perfectionism as a trait, but as a personality style or a pattern involving individual and social features…

A: Thank you for your suggestion. We revised the sentence and rephrased the statement.

  • R2: When perfectionism and intergenerational transmission is addressed, genetics must be mentioned. At least a reference about it should be included, and again mentioned in the discussion.

A: Thank you for the comment. We cited studies about genetics in the introduction, as requested (please see Iranzo-Tatay et al., 2015, and Tozzi et al., 2004).

  • R2: In the literature review of perfectionism and personality traits, studies with children are missing, and they should be included as children are the focus of this study. Namely, “Child Perfectionism and its Relationship with Personality, Excessive Parental Demands, Depressive Symptoms and Experience of Positive Emotions (Oros et al., 2017), “Clarifying the two facets of Self-Oriented Perfectionism: influences on affect and the Big Five traits of personality in children (Vicent et al., 2019).

A: Thank you for your suggestion. We included these studies in the introduction, as requested.

  • R2: P. 3, line 92: These studies about the relation of perfectionism with personality dimensions are with adults, and it should be mentioned. After that, studies with children should be introduced.

A: Thank you, we mentioned that those studies were carried out with adults, as kindly requested.

  • R2: Pertaining to assessment of the children´s personality, why did not the authors choose a self-report instrument, and instead they chose assessment by the parents? This must be explained in the paper, and it should come in the limitations too (see below).

A: Thank you for raising this point. We agree with this comment and discussed the use of parent reports of children’s personality as a limitation of the study, as follows:

Another limitation is the use of parent reports of children’s personality may be biased (e.g., some parents may have under-reported their children’s problematic personality attitudes) although they provide one external source of information that could be integrated with multimethod assessments in future studies”

  • R2: - P.4, linha 181,” In the present sample, we found alpha coefficients of the five domains ranging from .60 to .911, with high scores indicating high levels of the traits”. - This sentence seems to indicate that higher alfa values indicate higher trait levels…it should be reformulated.

A: Thank you. The sentence was modified according to your suggestion.

  • R2: P.4, line 171, The Hierarchical Personality Inventory for Children: the authors should present to which five general domains the facets belong; in the same vein, in the Discussion, when facets are presented, they should be discussed as framed in the respective general domain.

A: We agree with your suggestion and specified to which general domains the facets belong either in the Method and in the Discussion section

Results:

  • R2: The authors mention tables SI-     , it is incomplete, and I do not see any of these tables. A table with all the correlations should be presented. 

Thanks for noting. We indicated all the correlations analyses tested in Supplementary Materials.

  • R2: P.5, line 211, “Three separate hierarchical regression analyseswere performed to identify whether  mothers and fathers’ perfectionism (Step 1) predicted…”. - All the steps should be indicated. In p.8, line 299, they indicate step 2 only…these informations should be uniform.

A: Thank you for this comment. We integrated this information with other regression analysis results,  as follows:

“Regression analyses performed on mothers’s reports showed that variables entered in the Step 1 explained the 12% of the variance of SOP-Critical (R = .41, F(3, 53) = 3.36, p < .05), with mothers’ MPS-SPP (ß = .388, p < .05)  positively predicted children’s SOP-Critical. (…) Lastly, regression analyses performed on fathers indicated that variables entered in the Step 1 explained the 10% of the variance of CAPS-SPP (R = .39, F(3, 48) = 2.82, p < .05) ), with fathers’ MPS-SPP (ß = .434, p < .05)  positively predicted children’s CAPS-SPP.  Lastly, variables entered in the Step 2 explained 19% of the variance of CAPS-SPP (R = .57, F(3, 48) = 2.44, p < .05) and showed that Agreeableness (ß = -.355, p < .05), Conscientiousness  (ß = -.341, p < .05) and MPS-SPP (ß = .402, p = .05)  explained uniquely contributed to children’s CAPS-SPP.

  • R2: P.7, line 245, “T-tests comparing perfectionism (i.e., MPS) and personality (i.e., BFI-10) of parents of ID and ED children did not show any significant difference between these two groups of parents (all ps> .05). T-test comparing each scale of the MPS and of the BFI-10 filled in by mothers to those filled in by fathers were conducted separating ID and ED children. Results indicated that ED fathers reported higher levels of SOP (t = 2.19, p < .05) and higher levels of OOP (t = 2.16, p < .05) than ED mothers. No other significant differences were observed”. - I do not understand this paragraph. In the first sentence the authors state that there are no differences…Then, in the second sentence they mention the same variables of the first sentence, but state something different, and in the last sentence they state there are differences…so what does the first sentence refer to?

A: Thank you for this useful comment. The first sentence refers to the differences on MPS-S and BFI-10 between ED and ID parents, without considering their gender (e.g., whether there were differences between mothers and fathers). We confirm that there were no differences in this analysis.
The second sentence reported results from the analysis examining whether there were differences between mothers and fathers within the two groups of children. We tested this in each group of parents separately (ID parents and ED parents).

However, following the reviewer’s suggestion at question 7), these series of t-test were replaced with one MANOVA examining differences between ED and ID parents controlling for gender (mothers vs fathers) (see the comment below).

  • R2: T-test is to one dependent variable; to multiple dependent variables Manova should be used. Or, in alternative, Bonferroni correction should be used in order to counteract the multiple comparisons problem.

Thank you for raising this point. We agree with this comment and computed MANOVA for each comparison previously performed with T-tests. We specified this new procedure in the Statistical analysis section (A) and in the Results section (B):

  1. Multivariate analysis of variance was used for comparing children with ED and children with ED as well as the respective groups of parents on personality dimensions and perfectionism aspects.

  1. Perfectionism self-reported by ID and ED children
    One MANOVA comparing CAPS scales of children with ID and ED was performed, evidencing no significant differences in perfectionistic aspects between the two groups (all ps > .05).

Personality characteristics and perfectionism of the parents
With the aim of examine perfectionism and personality in ED and ID parents, one MANOVA was used to compare MPS-S and BFI scores reported by these two groups, controlling for gender (mothers vs fathers). Results indicated no significant effect for the group of diagnosis. However, a significant effect of gender was observed, with fathers presenting higher scores in MPS-SOP (M = 21.816; SD = 7.661; F(1, 100) = 5.207, p < .05; partial η2 = .05), higher MPS-OOP (M = 19.897, SD = 5.986; F(1, 100) = 7.561, p < .01; partial η2 = .07), and higher MPS-SPP (M = 17.795, SD = 7.424; F(1, 100) = 4.712, p < .05; partial η2 = .05) than mothers (SOP: M = 18.333, SD = 7.541; OOP: M = 16,333, SD = 6.796; SPP: M = 14.759, SD = 6.564).

Differences between ED and ID personality of children referred by their parents
In order to examine differences between ED and ID children’s personality referred by their parents, one MANOVA was computed considering parents groups (ED vs ID) as independent factors and controlling for gender (mothers vs fathers). Results evidenced that parents of children with ED reported higher levels of Creativity (M = 4.057; SD = 2.209; F(1, 100) = 4.184, p < .05; partial η2 = .04) and Perseverance (M = 4.80; SD = 2.083; F(1, 100) = 6.116, p < .05; partial η2 = .06) for their children, compared to parents of children with ID (respectively, M = 3.117; SD = 2.175 and M = 3.794; DS = 1.897). Moreover, parents of children with ID report higher levels of Obedience (M = 4.853; SD = 1.668; F(1, 100) = 5.422, p < .05; partial η2 = .05) in their children, compared with parents of children with ED (M = 3.137; SD = 1.141). No effects were observed for gender.

  • - P.7, line 253, “Since evaluations of children’s personality by mothers and fathers did not differ (as reported above), the t-test comparing personality traits of diagnostic groups of children (ID and ED) were conducted on the mean scores of both parents evaluations”. - Some paragraphs below the authors report data of mother and father separately, it is not coherent. In the Abstract this separated data is also present.

A: Thank you. As the statistical analysis approach was modified accordingly to the reviewer’s suggestions, this part was removed and replaced with results from MANOVA indicated above. We hope that the reviewer will be satisfied with this substantial improvement of the manuscript.

DISCUSSION

R2 : In general, I think it this section should be improved, clearer (by discussing the personality facets framed in the respective general domain, as I stated above) and more linked with theoretical aspects presented in the Introduction.

A: Thank you. We improved the discussion as suggested.

P.9, line 349, Agreeableness is repeated.

A: We have modified this sentence.

R2: Neuroticism did not have significant effect in any of the analyses; this must be addressed, as it is not expected by the literature.

A: We better explained the results about this section.

R2: The theoretical models that explain transgenerational transmission of perfectionism, which are presented in Introduction, should be included in the Discussion, addressing the results. As an example, when the authors state “These findings suggest not only that perfectionism of the parents may promote the development of perfectionistic tendencies in children but also that the influence may be different for mothers and fathers.”, this should be related with theoretical aspects presented in the Introduction.

A: We have included in this section the literature about the transgenerational transmission of perfectionism.

R2: Limitations

The report of personality characteristics of the children by the parents should be here, as there may be a bias that affects the relation between characteristics of the parents and reported characteristics of the children.

A: Thank you. We have discussed this point in the section Limitations.

Round 2

Reviewer 1 Report

The authors have done well to address my comments.